# Estrogen Enhances *FDFT1* Expression in Theca Cells of Chicken Hierarchical Ovarian Follicles by Increasing LSD1Ser54p Level Through GSK3β Phosphorylation at 216th Tyrosine

**DOI:** 10.3390/biom14111343

**Published:** 2024-10-22

**Authors:** Yanhong Zhang, Conghao Zhong, Xinmei Shu, Qingxin Liu, Yunliang Jiang

**Affiliations:** 1College of Life Science, Shandong Agricultural University, Tai’an 271018, China; 2022010140@sdau.edu.cn; 2College of Animal Science and Technology, China Agricultural University, Beijing 100193, China; chzhong@cau.edu.cn; 3College of Animal Science and Technology, Shandong Agricultural University, Tai’an 271017, China; 2023110348@sdau.edu.cn; 4Shandong Provincial Key Laboratory for Livestock Germplasm Innovation & Utilization, Shandong Agricultural University, Tai’an 271017, China

**Keywords:** chicken follicle, post-TCs, LSD1Ser54p, estrogen, GSK3βTyr216p, FDFT1

## Abstract

The development of chicken ovarian follicles involves two key stages of primordial follicle recruitment and follicle selection that are tightly regulated by multiple reproductive hormones and cytokines. Our previous study revealed an estrogen-stimulated increase in the phosphorylation level of serine at position 54 of lysine demethylase 1A (LSD1Ser54p) in the theca cells of chicken hierarchical ovarian follicles (Post-TCs). In this study, we further found that the upregulation of LSD1Ser54p by estrogen was performed by glycogen synthase kinase 3 beta (GSK3β) and that GSK3β promoted LSD1Ser54p levels by directly binding to the SWIRM and AOL1 domains of LSD1. Upon estrogen stimulation, the phosphorylation level of tyrosine at position 216 of GSK3β (GSK3βTyr216p) increased, which enhanced the binding between LSD1 and GSK3β. The subsequent transcriptome sequencing on chicken Post-TCs treated with estrogen and CUT&RUN sequencing against the LSD1Ser54p protein revealed that the expression of the farnesyl-diphosphate farnesyltransferase 1 (*FDFT1*) gene was simultaneously upregulated by estrogen, GSK3β, and LSD1Ser54p. Moreover, the overexpression of *FDFT1* further promoted cholesterol biosynthesis in chicken Post-TCs. In short, the findings of this study suggest that estrogen-induced tyrosine phosphorylation at position 216 of GSK3β can upregulate the level of LSD1Ser54p, leading to the activation of *FDFT1* expression and subsequently promoting cholesterol biosynthesis in chicken Post-TCs, which may in turn enhance estrogen synthesis.

## 1. Introduction

The normal development of ovarian follicles in the abdominal cavity of hens determines their egg laying performance, which is mainly divided into three stages: the stationary primordial follicle stage, the pre-hierarchical follicle stage in which follicles grow slowly but are prone to atresia, and the orderly hierarchical follicle stage [1,2]. Follicle selection occurs during the process of the transition from pre-hierarchical follicles (6–8 mm) to hierarchical follicles; the follicles being selected rapidly deliver yolk protein to the oocyte through a vascularized membrane layer [3]. The identification of factors controlling chicken follicle selection is important for improving egg production. Current investigations on the mechanism of chicken follicle selection mainly focus on transforming the growth factor beta family (TGF-β) and its hormonal regulation [4,5], and the role of estrogen in this process remains largely unknown.

Granulosa cells and theca cells are essential somatic components of chicken follicles, playing a crucial role in follicle development and hormone secretion [6]. Granulosa cells accelerate differentiation after follicle selection and receive follicle-stimulating hormone (FSH) stimulation to synthesize progesterone, which is crucial for maintaining reproductive hormone stability [7]. Both estrogen and progesterone are steroid-related hormones that require multi-step catalytic synthesis by using cholesterol as a substrate. In poultry species, estrogen is primarily secreted by follicular theca cells and influences reproductive functions by promoting granulosa cell proliferation [8,9], follicle development, and maturation [10,11,12], Understanding the molecular mechanisms underlying cholesterol biosynthesis during follicular development is essential for elucidating differences in egg laying performance among chicken breeds.

Initially isolated from HeLa cells, lysine demethylase 1A (LSD1) was found to be homologous to polyamine oxidase and interact with histone deacetylase HDAC2 [13]. LSD1 catalyzes the demethylation of lysine in the fourth position of the H3 histone (H3K4me1/me2) [14], involving the oxidation of the C-N bond of the substrate to generate amino groups, formaldehyde, and H_2_O_2_, which requires protons on the amine substrate to undergo the oxidation reaction. Unlike histone acetylation, histone methylation does not alter the net charge of chromatin but provides new docking sites for the recognition and binding of reading proteins [15]. LSD1 may act as a transcription inhibitor or activator by demethylating histones. During the process of red blood cell differentiation, LSD1 binds to 40 amino acid sequences in TAL1 (SCL), thereby inhibiting the target of TAL1 [16]. However, in vitro, ERRα induces LSD1 to demethylate H3K9, thereby enhancing the expression of matrix metalloproteinase 1 and increasing the ability of cells to invade the extracellular matrix [17]. In addition to histone H3, LSD1 can also demethylate non-histone proteins including TP53, E2F1, and DNMT1 [18,19,20], regulating their protein–protein interactions, activity, or stability.

Signal transduction within organisms typically involves kinases acting as mediators, transmitting external or internal signals to cells, enabling them to respond accordingly [21]. The kinase-mediated phosphorylation of LSD1 serine residues is reported to affect their function. PKCα-mediated LSD1 phosphorylation enhances the metastasis of breast cancer by regulating epithelial mesenchymal transformation [22]. The phosphorylation of LSD1 at Ser 126 by PLK1 supports chromatin release during mitosis [23]. GSK3β is a proline-directed serine/threonine kinase involved in various cellular pathways and diseases, including the metabolic and DNA repair pathways, cancer cell proliferation, and neurodegenerative diseases [24]. The GSK3β-dependent phosphorylation of LSD1 contributes to the stability of the LSD1 protein, and phosphorylated LSD1 enhances its binding affinity with ubiquitin-specific protease 22 (USP22) in glioma stem cells, thereby promoting tumorigenicity [25].

Our previous study revealed that LSD1Ser54p in the granulosa cells of chicken ovarian follicles is involved in chicken follicle selection [26] and is mediated by CDK5 [27]. In this study, we further characterized the role of LSD1Ser54p in chicken Post-TCs and found that estrogen treatment promotes GSK3βTyr216p, leading to its binding to LSD1, and causes a subsequent increase in the LSD1Ser54p level. Moreover, we found that, as a key enzyme in steroid biosynthesis [28], *FDFT1* is upregulated by estrogen stimulation through GSK3βTyr216p and LSD1Ser54p. Collectively, we demonstrated for the first time that estrogen stimulates *FDFT1* expression and cholesterol synthesis by upregulating LSD1ser54p through GSK3βTyr216p in chicken Post-TCs.

## 2. Methods

### 2.1. Animals and Sample Collections

Forty-eight Hy-line brown laying hens aged 35 to 40 weeks were used to collect ovarian follicles; the hens are kept separately in egg laying cages, with free access to food and water. They are given 16 hours of light and 8 hours of darkness, and their egg laying time is recorded. Approximately 10 h after laying, the hens were slaughtered by cervical dislocation, and from each hen, hierarchical follicles including F6 to F1 were collected and washed with phosphate-balanced solution (PBS). Animal experiments were authorized by the Institutional Animal Care and Use Ethics Committee of Shandong Agricultural University (No. SDAUA-2021-097) and performed in accordance with the “Guidelines for Experimental Animals” outlined by the Ministry of Science and Technology of China.

### 2.2. Cell Culture and Treatment

Chicken Post-TCs were isolated and cultured according to reference [29]. Briefly, tweezers were used to remove the outer matrix carefully and gently squeeze egg yolk out of chicken hierarchical follicles, the remaining outer membrane of which was cleaned with PBS, cut into pieces, and digested with 1% collagenase II at 37 °C (Coolaber, Beijing, China). After filtration and centrifugation, M199 medium (Gibco, Camarillo, CA, USA) containing 5% fetal bovine serum (Biological Industries, Kibbutz Beit Haemek, Israel) and 1% penicillin/streptomycin (Solarbio, Beijing, China) was used to culture these cells andfor 24 hours in a 6-well culture plate, and then treated the cells with estradiol (Sigma, Saint Louis, MO, USA) and CHIR99021 (Santa Cruz Biotechnology, Inc., Dallas, TX, USA). DMEM (Gibco, Camario, CA, USA) supplemented with 10% fetal bovine serum was used to culture 293T cells (Passage Number < 30) in a 37 °C incubator (Thermo Fisher Scientific, Waltham, MA, USA) and 5% CO_2_.

### 2.3. Plasmid Construction and Cell Transfection

Total RNA was extracted out of theca cells using an RNA isolation kit (TIANGEN, Beijing, China), and the quality and concentration were checked using spectrophotometry (Eppendorf, Hamburg, Germany) and 1% agarose gel electrophoresis. The synthesis of complementary DNA (cDNA) was carried out via reverse transcription utilizing All-in-one First-Strand Synthesis MasterMix (Yugong Biotech, Jiangsu, China). Then, 2 × Phanta Max Master Mix (Vazyme, Nanjing, China) was used for amplifying DNA fragments according to the following program: at 95 °C for 3 m, then 35 rounds of 95 °C denaturation for 15 s, 60 °C for 15 s, 72 °C for 1 m 30 s, and finally 72 °C for 5 m. A homologous recombination kit (Vazyme, Nanjing, China) was used to clone PCR products into the pcDNA3.1-Fg/Myc vector. Using wild-type plasmids as templates, the LSD1S54A and GSK3βT216A mutant plasmids were subsequently amplified. All primers are listed in Appendix A. When the cell density reached 70 to 80% confluence on 6-well plates, the LTX Plus Agent reagent (Invitrogen, Carlsbad, CA, USA) and PEI reagent (Proteintech Group, Wuhan, China) were used to transfect chicken Post-TCs and 293T cells for 6 h, respectively, according to the manufacturer’s instructions; then, they were replaced with the corresponding complete culture medium and incubated for an additional 24 h or 36 h. For each well, 2 μg of plasmid was used.

### 2.4. Protein Isolation and Western Blotting

Lysis buffer (NCM, Suzhou, China) containing a phosphoprotease inhibitor (Beyotime, Shanghai, China) was used to treat cells at 4 °C for 30 min; the protein was obtained by centrifugation at 4 °C and 12,000 rpm for 10 min, the concentration of which was measured with an enhanced BCA detection kit (Beyotime, Beijing, China). Proteins were separated by electrophoresis on a 4–15% BeyoGelTM Plus prefabricated PAGE (Beyotime, Beijing, China) and then transferred to a PVDF membrane (Roche, Mannheim, Germany). At room temperature, the PVDF membrane (New Cell & Molecular Biotech, Suzhou, China) was sealed for 15 min and washed three times for 10 min each time in TBST (Coolaber, Beijing, China), incubated overnight with primary antibody at 4 °C, and rewashed three times using TBST. The primary antibodies used in this experiment are Myc (ABclonal, Wuhan, China), Flag (ABclonal, Wuhan, China), β-actin (Beyotime, Beijing, China), GAPDH (Beyotime, Beijing, China), LSD1 (ABclonal, Wuhan, China), LSD1Ser54p (ABclonal, Wuhan, China), GSK3β (ABclonal, Wuhan, China), and GSK3βTyr216p (ABclonal, Wuhan, China). The dilution ratio of the Myc, Flag, β-actin, and GAPDH antibodies is 1:4000, and the dilution ratio of the LSD1, LSD1Ser54p, GSK3β, and GSK3βTyr216p antibodies is 1:2000. Then, the PVDF membrane was incubated with goat anti-rabbit IgG (H+L) (Beyotime, Beijing, China) or goat anti-mouse IgG (H+L) (Beyotime, Beijing, China) diluted to 1:8000 at room temperature for 2 h and washed three times with TBST. The C300 imaging system (Azure Biosystems, Dublin, CA, USA) and the chemiluminescence liquid (NCM, Suzhou, China) were used to visualize the imprint signals on PVDF membranes. Image J (v1.4.3) software was used to evaluate protein levels by quantitatively dividing the grayscale intensity of the target protein by the grayscale intensity of the reference protein. Three biological replicates were performed for the expression analysis of each protein. Original Western blots can be found at Appendix A.

### 2.5. Immunofluorescence

Cells were plated onto cell slides, allowed to culture for 24 h, and then washed with PBS three times. Fixation was performed by treatment with 4% paraformaldehyde (Biosharp, Guangzhou, China) for 10 min and then washing three times with PBS; then, 0.5% Triton X-100 (Blotopped, Beijing, China) was added and treated for 15 min and washed three times with PBS. The slides were sealed for 30 min with 10% goat serum (SL038, Solarbio, Beijing, China), incubated with the GSK3β and GSK3βTyr216p primary antibodies (ABclonal, Wuhan, China) (diluted 1:50) at 37 °C for 1 h, washed with PBS three times, then incubated with a homologous fluorescent secondary antibody (h+L) (Beyotime, Shanghai, China) in the dark at 37 °C or 1 h, and finally washed with PBS three times. DAPI (Solarbio, Beijing, China) was used to stain cell nuclei at room temperature for 3–5 min. A fluorescence microscope (Leica, Wetzlar, Germany) was used to observe cells.

### 2.6. RNA Isolation and Real-Time Quantitative PCR

Total RNA from chicken Post-TCs was extracted as described in Section 2.3. Using total RNA as a template, cDNA was synthesized with All-in-one First-Strand Synthesis Master Mix (Yugong Biolabs Co., Ltd.). Genomic DNA was removed by incubating the mixture of RNA and the genomic DNA remover at 37 °C for 2 min; afterward, the reverse transcription mixture was added to the mixture and further incubated at 50 °C for 15 min and 85 °C for 5 min to obtain cDNA. The SYBR Premix Ex Taq ™II kit (TaKaRa, Dalian, China) was used for the real-time quantitative PCR (qPCR) detection of the *NR5A1*, *BMPR1β*, *SMAD2*, *CDKN2B*, *HSD3β1*, *CYP11A1*, *STAR*, *CYP24A1*, *FDFT1*, and *LSD1* mRNA expression levels using the primers listed in Appendix A, and the housekeeping gene *GAPDH* was used as the reference. The following program was used on the Light Cycler 96 real-time PCR system (Roche, Basel, Switzerland): 95 °C for 30 s, followed by 40 cycles of 95 °C denaturation for 10 s, annealing for 30 s, and 72 °C for 30 s. The melting curves were obtained, and quantitative data analysis was performed using the 2^−ΔΔCT^ relative quantification method [30].

### 2.7. Immunoprecipitation

Chicken Post-TCs were treated with Cold RIPA lysis buffer (Beyotime, Shanghai, China) with a complete protease inhibitor (NCM, Suzhou, China) to obtain the total protein that was incubated with 1 to 2 mg of the Flag (Engibody, Beijing, China), Myc (Engibody, Beijing, China), LSD1 (ABclonal, Wuhan, China), and GSK3β (ABclonal, Wuhan, China) primary antibodies at 4 °C for 4 h, and subsequently, 20 μL of Protein A/G PLUS-Agarose beads (Engibody, Beijing, China) was added and incubated at 4 °C for 2 h with rotation. Immunocomplexes were extensively washed four times with PBS, heated to 100 °C for 10 min, and separated by SDS-PAGE. Three biological replicates were performed for each immunoprecipitation test.

### 2.8. Purification of Nuclear and Cytoplasmic Proteins

The preparation and purification of nuclear protein and cytoplasmic protein from chicken Post-TCs were carried out using a nuclear protein and cytoplasmic protein extraction kit (Beyotime, Shanghai, China), and the extraction process was carried out according to the supplier’s instructions. A BCA assay kit (Beyotime, Shanghai, China) was used for protein quantification and denaturation extraction. The sample was stored at −20 °C and then used for protein blotting.

### 2.9. Transcriptome Analysis

Total RNA was extracted from chicken Post-TCs using an RNA isolation kit (Tiangen, Beijing, China). The quality and concentration of the RNA were assessed with the Agilent 2100 Bioanalyzer (Agilent Technologies, Santa Clara, CA, USA) and used for mRNA library construction (2 groups, n = 3 for each). The Ultra TM RNA Library Prep Kit was used to generate sequencing libraries, and sequencing was performed on the Illumina Novaseq platform. The raw data underwent filtering and quality control to produce clean data. The index of the reference genome was built, and the paired-end clean reads were aligned to the Gallus gallus genome (https://ftp.ncbi.nlm.nih.gov/genomes/all/GCF/000/002/315/GCF_000002315.5_GRCg6a/, accessed on 7 January 2020) using Hisat2 v2.0.5 [31]. The FPKM method was used to quantify gene expression levels [32], with screening criteria for differentially expressed genes (DEGs) set at |log2 (Fold Change)| > 1 and *p* value < 0.05, comparing conditions before and after estrogen treatment. The Gene Ontology database was used to analyze the differentially expressed mRNAs. Pathway analysis was carried out using the KEGG database (http://www.kegg.jp/kegg) to elucidate the biological pathways enriched among the differentially expressed mRNAs [33,34,35].

### 2.10. Cut&Run qPCR and Cut&Run Sequencing

In chicken Post-TCs, the Hyperactive pG MNase CUT&RUN Assay Kit for Illumina (Vazyme, Nanjing, China) was used to obtain DNA fragments targeted by LSD1Ser54p following the manufacturer’s instructions. The obtained DNA was subjected to qPCR analysis and sequencing; qPCR detection was performed according to the steps in Section 2.6. The DNA samples from the library were sequenced on the Illumina HiSeq platform, obtaining raw data in FASTQ format. The raw data of each sample were separately counted, including the sample name, read number, base number, percentage of fuzzy bases (undetermined bases), and Q20 (%) and Q30 (%). The sequencing data were filtered as follows: (1) Fastp was used to remove DNA sequences with 3’ end adapters, and (2) low quality sequences were removed. An index of the reference genome was established, and the paired-end clean readings were compared with the *Gallus gallus* genome (Gallus_gallus.bGal1.mat.browser.GRCg7b.dna.toplevel) (http://asia.ensembl.org/Gallus_gallus/Info/Index, accessed on 10 September 2021). The genome was annotated to obtain positional, taxonomic, and descriptive annotations for each gene, which were used for the subsequent classification, enrichment, and clustering analyses. MACS2 (v2.2.6) software was used for Call Peak, with a filtering threshold q value < 0.05. Then, the significantly enriched regions (peaks) on the reference genome alignment were calculated. After Peak Calling, the signal values of each locus on the genome were obtained. The average signal values of all genes at each locus within a range of 3 kb upstream of the transcription start site (TSS) to 3 kb downstream of the transcription end site were calculated. According to the results of Peak Calling, the number of common and unique DEGs in each gene group was compared. The find Motifs Genome, Pl tool of the HOMER (v4.10) software was used to predict the motifs of these DNA sequences. Then, the predicted motifs were matched with existing motif data in the databases HOMER and JASPAR. ChIPseeker was used to annotate genes located in the peak region and nearby genes, obtaining relevant information on their functions and metabolic pathways.

### 2.11. ELISA

The chicken total cholesterol enzyme-linked immunosorbent assay kit (MEIMIAN, Yancheng, China) was used to analyze cholesterol secretion in chicken Post-TCs according to the manufacturer’s instructions. After the reaction was terminated, the OD values of each well were measured at a wavelength of 450 nm (Eppendorf, Hamburg, Germany). A standard curve was plotted with the standard substance concentration as the horizontal axis and the OD value as the vertical axis. Then, based on the OD value of the sample to be tested, the corresponding concentration was calculated using the standard curve formula, and it was multiplied by the dilution factor to obtain the actual concentration of the sample.

### 2.12. Statistical Analysis

All experiments were repeated at least three times, and all data are presented as the mean ± SEM. Differences between the experimental groups were evaluated by a one-way ANOVA followed by Duncan’s multiple range test (*p* < 0.05) using the General Linear Model procedure of IBM SPSS Statistics 20. Data plotting was performed by GraphPad Prism software (version 8.0; San Diego, CA, USA).

## 3. Results

### 3.1. GSK3β Is the Kinase of LSD1Ser54p in Chicken Post-TCs

Our previous study showed that, after follicle selection, the level of LSD1Ser54p increased in chicken hierarchical granulosa cells [26], and GSK3β was predicted to perform this phosphorylation [27]. Herein, wild-type LSD1 and LSD1S54A mutants were overexpressed in 293T cells, and protein signals were detected using the LSD1Ser54p antibody that was specifically prepared. A strong LSD1Ser54p protein signal was detected in the group expressing wild-type LSD1, rather than in the group expressing the LSD1 mutant (Figure 1A), indicating the high specificity of the customized antibody.

After the overexpression of GSK3β in Post-TCs, the LSD1Ser54p level significantly increased, while the LSD1 level did not change significantly (Figure 1B). When chicken Post-TCs were treated with CHIR99021, a GSK3β inhibitor, the level of LSD1Ser54p was reduced (Figure 1C), suggesting that GSK3β is the kinase of LSD1Ser54p in chicken Post-TCs.

### 3.2. Direct Interaction Between LSD1 and GSK3β

To confirm that LSD1Ser54p was directly phosphorylated by GSK3β, a co-immunoprecipitation assay was used to detect the potential interaction between GSK3β and LSD1. LSD1-Myc and GSK3β-Fg were coexpressed in 293T cells, and LSD1-Myc was found to be present in the anti-FLAG immune complex and vice versa (Figure 2A). Similarly, in chicken Post-TCs, endogenous LSD1 was present in the endogenous GSK3β immune complex and vice versa (Figure 2B).

To further analyze this interaction, we generated LSD1 functional domain truncation mutants, namely N+SWIRM-Myc (1–246) containing protein interaction domains and ATA-Myc (247–827) containing two substrate binding domains, as well as two separate AOL domains: AOL1-Myc (247–389) and AOL2-Myc (490–827). The results showed that N+SWIRM-Myc (1–246), ATA-Myc (247–827), and AOL1-Myc (247–389) were present in the immune complex of GSK3β-Fg, while AOL2-Myc (490–827) was not (Figure 2C).

We further investigated the localization of GSK3β in chicken Post-TCs and found that GSK3β was detected in both the cytoplasm and nucleus, but it was mostly localized in the nucleus (Figure 2D). Our previous study revealed that LSD1Ser54p was also localized in chicken Post-TCs. These results indicate that GSK3β directly interacts with LSD1, and this interaction requires the participation of the SWIRM and AOL1 domains.

### 3.3. Phosphorylation of GSK3β at 216th Tyrosine Upregulates LSD1Ser54p Level upon Estrogen Treatment

Studies have shown that the phosphorylation of serine at position 9 mainly inhibits its kinase activity, while the phosphorylation of tyrosine at position 216 promotes its activity [36]. Phosphorylation modifications at these two sites can affect the protein spatial structure of GSK3β and further affect its binding to substrates. To test the effect of tyrosine phosphorylation at position 216 of GSK3β on LSD1Ser54p, a mutant plasmid with alanine at position 216 of GSK3β was generated. After overexpression in chicken Post-TCs, it was found that LSD1Ser54p and LSD1 level did not change (Figure 3A). LSD1-Myc and GSK3βT216A-Fg were coexpressed in 293T cells, and the co-immunoprecipitation results showed that the level of binding of LSD1 to GSK3βT216A was significantly reduced compared to that of wild-type GSK3β (Figure 3B).

Our previous study found that estrogen increased the level of LSD1Ser54p in chicken Post-TCs [26]. We further treated chicken Post-TCs with 50 ng/mL of estrogen and found that the phosphorylation level of tyrosine at position 216 of GSK3β (GSK3βTyr216p) started to increase 5 min after estrogen treatment, reached its maximum value at 30 min, and then decreased, while the LSD1Ser54p level increased after 30 min of treatment and decreased after reaching its maximum value at 1 h (Figure 3C). These results indicate that the phosphorylation of tyrosine at position 216 of GSK3β is essential for promoting the level of LSD1Ser54p.

We further examined the localization of GSK3βTyr216p in chicken Post-TCs and found that it was distributed in both the cell cytoplasm and nucleus but mainly in the nucleus (Figure 3D), consistent with the cellular localization of LSD1Ser54p, which was also confirmed by the subsequent nuclear cytoplasmic separation test (Figure 3E).

### 3.4. Changes in Transcription Profile of Chicken Post-TCs After Estrogen Treatment

Estrogen is synthesized in chicken Post-TCs and promotes follicle development and maturation. To identify genes regulated by estrogen, RNA-seq on chicken Post-TCs treated with estrogen was performed. After the quality control of the total sequencing reads, the clean reads obtained accounted for over 99.08% of the total, with a comparable rate of over 91.5% with the reference genome (Appendix A).

Based on the criteria of |log_2_ (Fold Change)| > 1 and *p* value < 0.05, a total of 1283 differentially expressed genes (DEGs) were identified (720 upregulated, 563 downregulated) (Figure 4A and Appendix A), and a hierarchical clustering diagram of DEGs was constructed (Figure 4B). In addition, as LSD1 often appears as a transcription cofactor, the expression changes in transcription factors after estrogen treatment were identified. The top 20 transcription factor families are shown in Figure 4C, with specific differential transcription factors being listed in Appendix A. KEGG analysis revealed the top three pathways with high enrichment levels of Mannose type O-glycan biosynthesis, ECM receptor interaction, and oocyte meiosis (Figure 4D). The mRNA expression of eight DEGs, including *NR5A1*, *BMPR1β*, *SMAD2*, *CDKN2B*, *HSD3β1*, *CYP11A1*, *STAR*, and *CYP24A1*, was analyzed by qPCR (Figure 4E), the results showed that the gene expression levels after estrogen treatment were similar to those of RNA-seq.

### 3.5. CUT&RUN Sequencing Analysis of LSD1Ser65p Directly Targeted Genes

To identify the directly targeted genes of LSD1Ser54p in chicken Post-TCs, customized antibodies specific to LSD1Ser54p were used for CUT&RUN sequencing, with two sequencing samples, pLSD1-1 and pLSD1-2, and one IgG group being used as a negative control. Statistics on the raw data of each sample, including the number of reads, number of bases, Q30, percentage of unannotated bases, and Q20 (%) and Q30 (%), are shown in Appendix A, and basic information on data filtering can be found in Appendix A.

According to the results of Peak Calling, 7935 overlapping genes were detected in the two samples (Figure 5A), which were mainly enriched in the promoter and intron regions (Figure 5B). A strong signal peak appeared near the TSS, indicating that peaks were mainly enriched near the TSS (Figure 5C). Most of the annotated genes were found to be enriched in the signal transduction, immune, and endocrine signaling pathways (Figure 5D) and participate in cellular processes and metabolism in biological processes, express cellular anatomical entities in cellular components, and play a binding role in molecular functions (Figure 5E).

### 3.6. Estrogen/GSK3β/LSD1Ser54p Coregulate the Expression of FDFT1

To elucidate the mechanism by which estrogen regulates chicken follicular development through LSD1Ser54p, the DEGs in chicken Post-TCs treated with estrogen and the LSD1Ser54p-targeted genes annotated by CUT&RUN sequencing were jointly screened. Genes with low expression levels were removed, and 471 candidate genes were identified (Figure 6A, Appendix A), among which 18 genes were related to follicle development, i.e., *HSD17β7*, *SC5D*, *HSD3β1*, *SQLE*, *CYP11A1*, *HSD17β12*, *DHCR7*, *MSMO1*, *CYP17A1*, *HMGCS1*, *DHCR24*, *FDFT1*, *NSDHL*, *CYP51A1*, *HSPA2*, *CACYBP*, *MGST1*, and *TMSB4X*. After the overexpression of wild-type LSD1 and LSD1S54A mutants in chicken Post-TCs, we found that the expression of *FDFT1* increased when overexpressing wild-type LSD1, while no significant change was detected after overexpressing LSD1S54A (Figure 6B, Appendix A).

*FDFT1* is a key rate-limiting enzyme in the process of steroid biosynthesis, especially affecting the cholesterol level. Cholesterol is a precursor substance for steroid-related hormones (estrogen, progesterone) during follicular development. The CUT&RUN sequencing results showed that the direct binding region of LSD1Ser54p was located at 2809 bp to 3687 bp of the chicken *FDFT1* gene, totaling 879 bp. This fragment includes 192 bp upstream of the transcription start site, 65 bp of 5′UTR, 102 bp of the first exon, and 520 bp of the intron. All these four segments had the highest enrichment in LSD1Ser54p, while there was no significant change in wild-type LSD1 (Figure 6C). Consistent with RNA-seq, the expression of *FDFT1* significantly increased after the estrogen treatment of chicken Post-TCs (Figure 6D). After treatment with the GSK3β inhibitor, CHIR99021, *FDFT1* was inhibited, consistent with the changes in LSD1Ser54 after treatment with CHIR99021 (Figure 6E). The four segments of *FDFT1* were mainly enriched in histone H3K9me1 and H3K4me2 modifications (Figure 6F,G). The motifs of LSD1Ser54p binding to the *FDFT1* segment are listed in Appendix A. These data indicate that *FDFT1* was a target gene jointly regulated by estrogen, GSK3β, and LSD1Ser54p.

### 3.7. FDFT1 Promotes Cholesterol Production in Chicken Post-TCs

The overexpression of *FDFT1* in chicken Post-TCs significantly increased cholesterol production (Figure 7A). After treatment with YM-53601, an enzyme activity inhibitor of *FDFT1*, cholesterol levels significantly decreased from 12 h to 36 h (Figure 7B). These results indicate that *FDFT1* plays a critical role in promoting cholesterol production in chicken Post-TCs.

## 4. Discussion

In poultry, estrogen promotes the development and activation of chicken primordial follicles, involving their increased production of receptors and cadherin in early ovarian development [37]. Studies have shown that estrogen typically binds to nuclear receptors (ERα, ERβ) and G protein-coupled receptors [38]; these receptors are linked to other DNA-binding transcription factors to form transcriptional regulatory complexes, which then activate target genes [39]. In addition to the well-studied transcriptional effects of estrogen, there is also a rapid effect that occurs within seconds or minutes after the addition of estrogen [40]. These rapid effects include the activation of kinases and phosphatases, as well as an increase in transmembrane ion flux. Although these rapid effects have been widely analyzed, there is still no consensus on whether classical estrogen receptors are involved or whether unique membrane-associated receptors are present [41,42,43].

GSK3β is an important cellular signaling protein that plays a crucial role in various biological processes, including cell proliferation, differentiation, and apoptosis. The phosphorylation state of GSK3β usually alters the activity of GSK3β, thereby affecting the phosphorylation state of downstream signaling molecules or the protein level of transcription factors, thereby regulating the biological function of target cells [44,45,46]. In this study, we revealed the rapid role of estrogen in regulating GSK3β activity. We found that treating chicken Post-TCs with estrogen induced the phosphorylation of tyrosine at position 216 of GSK3β within a few minutes (Figure 3C), suggesting that estrogen regulates multiple cellular signaling pathways by affecting the activity of GSK3β, thereby affecting cell fate determination and functional expression. The names and functions of the signaling molecules, proteins, and genes involved in this study are listed in Appendix A.

The amino acid residues of LSD1 are regulated by various kinases and post-translational modifications, impacting diverse cellular processes. Our previous study investigated the phosphorylation of LSD1 and LSD1Ser54p and found that in granulosa cells, the kinase regulating LSD1Ser54p is CDK5 and that the phosphorylation of this site enhances the activity of LSD1’s demethylase activity [27]. In this study, we analyzed the regulation of LSD1Ser54p in chicken Post-TCs, identified GSK3β as the kinase involved (Figure 1B,C), and revealed that the tyrosine at position 216 of GSK3β was used to mediate the phosphorylation of serine at position 54 of LSD1. It is reported that the phosphorylation of tyrosine at position 216 of GSK3β leads to an enhanced kinase activity of GSK3β [47], and specifically, the phosphorylation of the GSK3β residue Tyr 216 leads to a wider catalytic cell, thereby promoting substrate entry [48]. This is consistent with the present study, showing that the mutation of the tyrosine at position 216 of GSK3β to alanine significantly reduced the binding between the mutant and LSD1 (Figure 3B). It is predicted that the loss of phosphorylation ability by the GSK3β mutant leads to the narrowing of the catalytic tank, which prevents the entry of LSD1. The tissue-specific regulation of LSD1Ser54p by kinases in chicken granulosa cells and Post-TCs expanded our understanding of the regulation mechanism of LSD1’s activity. Further study is required to explore the changes in different cell types and under different physiological states in chicken ovarian follicles to gain a more comprehensive understanding of their biological significance.

After binding to DNA, transcription factors recruit co-activators or co-repressors, partially composed of chromatin-modifying enzymes that may promote or inhibit transcription through covalent modifications of histone tails [49]. LSD1 is involved as a modifying enzyme that regulates histone demethylation. Physical interaction between LSD1/CoREST and transcription regulatory factors is reported to promote inhibition, including the binding of LSD1 to the proline-rich domain of transcription factor PRDM1 (BLIMP1), which helps to inhibit mature B-cell gene expression programs during plasma cell differentiation (including regulating *CIITA*, *PAX5*, and *SPIB*) [50]. Similarly, LSD1 has been shown to function as a transcriptional co-activator; LSD1 is recruited to *UCP1* and other BAT-rich genes (such as *PGC1a*) by interacting with Zfp516 and by demethylating H3K9 [51]. In this study, we identified the direct binding target gene *FDFT1* of LSD1Ser54p and verified that LSD1Ser54p activated *FDFT1* transcription (Figure 6D). Given that the upregulation of LSD1Ser54p increases the activity of its own histone demethylase and that the demethylation of H3K9me1/2 occurs in the transcriptional active region, we predict that the transcriptional regulation of *FDFT1* is likely to be achieved through the H3K9 demethylase function of LSD1.

The molecular mechanism of *FDFT1* gene expression has been investigated in the liver. The component of psoralen (FP) in traditional Chinese medicine Psoralea can inhibit *FDFT1* through the AKT/mTOR/SREBP-2 pathway, lipid accumulation, and cholesterol synthesis [52]. Cellular cholesterol synthesis is regulated by sterol regulatory element-binding protein (SREBP) that participates in cholesterol synthesis by sensing cholesterol levels and activating gene transcription involved in lipid synthesis [53]. *FDFT1* is a target gene of SREBP [54]. It was demonstrated that LSD1 affects lipid metabolism by regulating the gene expression and protein stability of SREBP [55]. Further investigation is needed to determine if SREBP is involved in the transcriptional activation of *FDFT1* by LSD1Ser54p.

Squalene is a specific precursor for the synthesis of cholesterol. Encoded by *FDFT1*, squalene synthase (SQS) is an evolutionarily conserved enzyme that catalyzes the condensation of naphthyl diphosphate molecules to form squalene through presqualene diphosphate ester [56,57]. In humans, mutations in the *FDFT1* gene have a negative impact on cholesterol biogenesis, leading to a decrease in plasma cholesterol levels [58]. In this study, we found that the treatment of chicken Post-TCs with *FDFT1* inhibitors significantly reduced the cholesterol secretion of *FDFT1*, consistent with the function of *FDFT1* in promoting cholesterol production (Figure 7B). This study fills the gap in the molecular mechanism of cholesterol biosynthesis in poultry follicular Post-TCs, showing that estrogen promotes cholesterol synthesis by promoting *FDFT1* gene expression. Whether a similar mechanism is used to regulate cholesterol biosynthesis in the granulosa cells of chicken ovarian follicles needs to be further investigated.

## 5. Conclusions

In summary, in chicken theca cells of hierarchical follicles, estrogen promotes tyrosine phosphorylation at position 216 of GSK3β to activate GSK3β kinase activity and then upregulates the level of LSD1Ser54p, which activates the mRNA expression of *FDFT1* and enhances cholesterol production (Figure 8). These findings unveil a novel mechanism by which estrogen regulates the synthesis of cholesterol in chicken hierarchical follicles, which is beneficial for understanding the mechanisms regulating cholesterol synthesis in the ovarian follicles of chickens as well as in mammalian species.

## Figures and Tables

**Figure 1 biomolecules-14-01343-f001:**
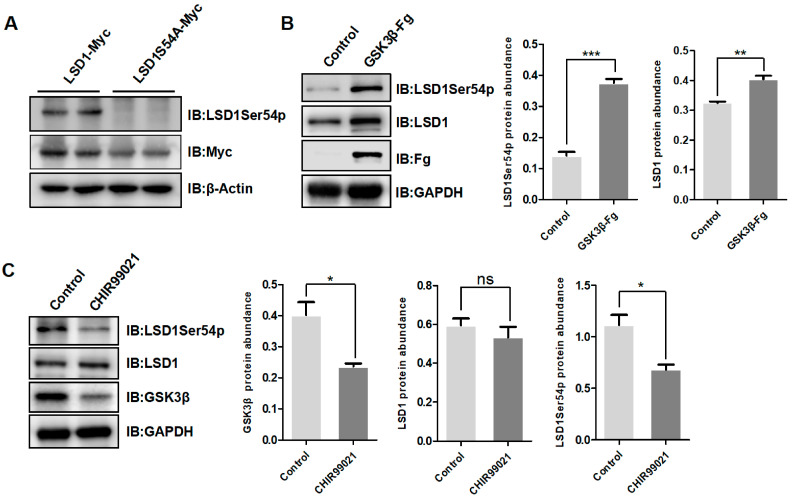
Effect of GSK3β on LSD1Ser54p expression. (**A**) Western blotting of LSD1Ser54p protein after overexpression of wild-type LSD1 and LSD1S54A in 293T cells. (**B**) Western blotting of LSD1Ser54p after overexpression of GSK3β in chicken Post-TCs. Statistical results of protein grayscale values are on right. (**C**) Western blotting of LSD1Ser54p in chicken Post-TCs treated with GSK3β inhibitor CHIR99021 (5 μmol). Statistical results of protein grayscale values are on right. * *p* < 0.05, ** *p* < 0.01, *** *p* < 0.001, ^ns^
*p* > 0.05.

**Figure 2 biomolecules-14-01343-f002:**
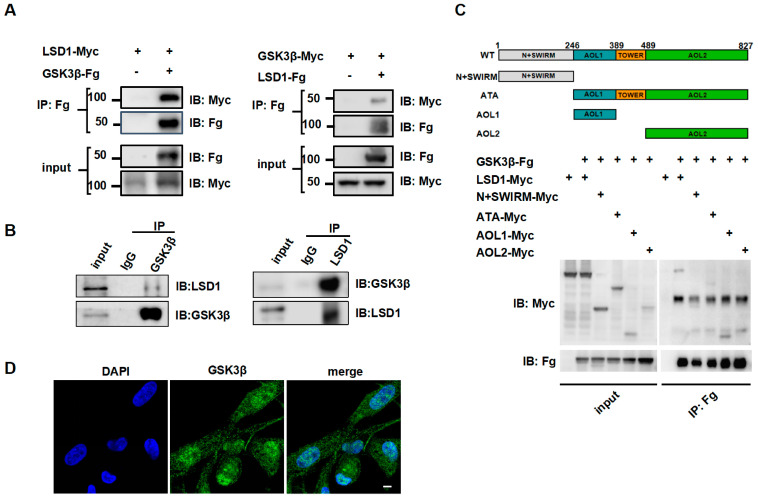
Co-immunoprecipitation assay of LSD1Ser54p and GSK3β. (**A**) 293T cells were transiently transfected with fusion plasmids of LSD1 and GSK3β. After 48 h, total cell lysis products were collected for co-immunoprecipitation and Western blotting. (**B**) Co-immunoprecipitation on total cell lysate of chicken Post-TCs and then Western blotting with corresponding antibodies were conducted. (**C**) Binding region of LSD1 with GSK3β in 293T cells. LSD1 encodes 827 amino acids. Its N-terminal contains protein interaction domain SWIRM (1–246), as well as two histone substrate-binding domains AOL1 (247–389) and AOL2 (490–827), and between these two domains, there is also TOWER (390–489) domain. The wild-type LSD1 plasmid and LSD1 functional domain truncated plasmid were transfected with GSK3β plasmid into 293T cells. After 48 h, total cell lysis products were collected for co-immunoprecipitation and Western blotting. (**D**) Immunofluorescence detection of localization of GSK3β in chicken Post-TCs. GSK3β, green. DAPI, blue. Scale bar, 15 μm.

**Figure 3 biomolecules-14-01343-f003:**
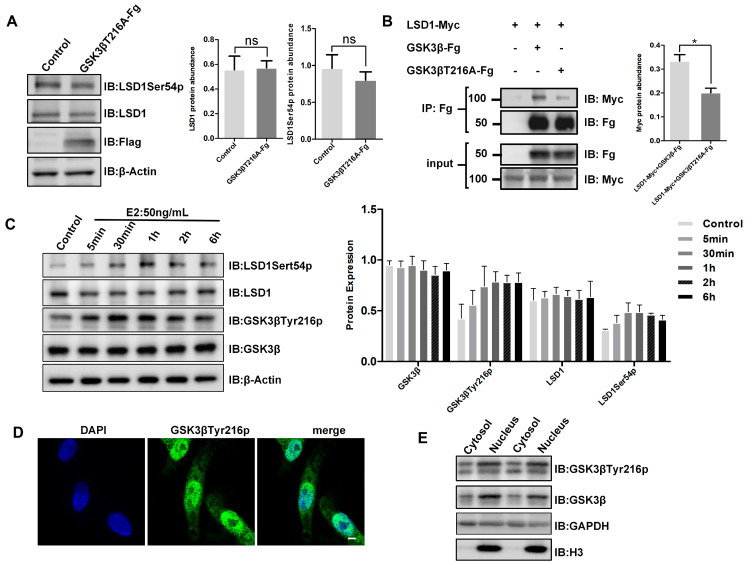
Estrogen regulates LSD1Ser54p protein levels through GSK3β. (**A**) GSK3βT216A mutant plasmid was transfected into chicken Post-TCs, and after 48 h, total cell lysate was extracted for Western blotting. On right are statistical results of protein grayscale values. (**B**) LSD1, GSK3β, and GSK3βT216A plasmids were transfected into 293T cells. After 48 h, total cell lysate was collected for co-immunoprecipitation and Western blotting. (**C**) Chicken Post-TCs were treated with 50 ng/mL estrogen for 5 min, 30 min, 1 h, 2 h, and 6 h. Total cell lysate was collected for Western blotting. On right are statistical results of protein grayscale values. (**D**) Immunofluorescence detection of localization of GSK3βTyr216p in chicken Post-TCs. GSK3βTyr216p, green; DAPI, blue; scale bar, 15 μm. (**E**) Western blotting for nuclear protein and cytoplasmic protein from chicken Post-TCs. * *p* < 0.05, ^ns^
*p* > 0.05.

**Figure 4 biomolecules-14-01343-f004:**
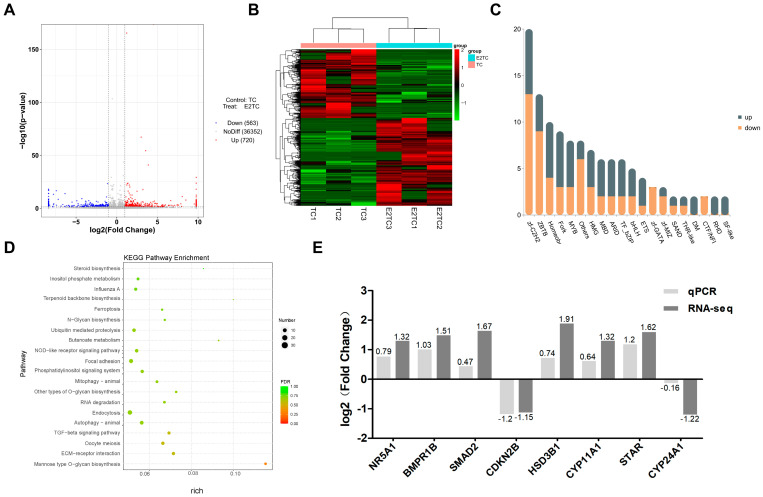
Changes in transcriptome profiles of chicken Post-TCs after treatment with 50 ng/mL estrogen. (**A**) Volcano plot of DEGs, with blue indicating downregulation and red indicating upregulation. (**B**) DEG clustering heat map. (**C**) Name and quantity of differential transcription factor family. (**D**) KEGG signaling pathway enriched with DEGs. (**E**) Changes in mRNA expression of eight randomly selected genes in chicken Post-TCs after estrogen treatment, as shown by qPCR and RNA-seq.

**Figure 5 biomolecules-14-01343-f005:**
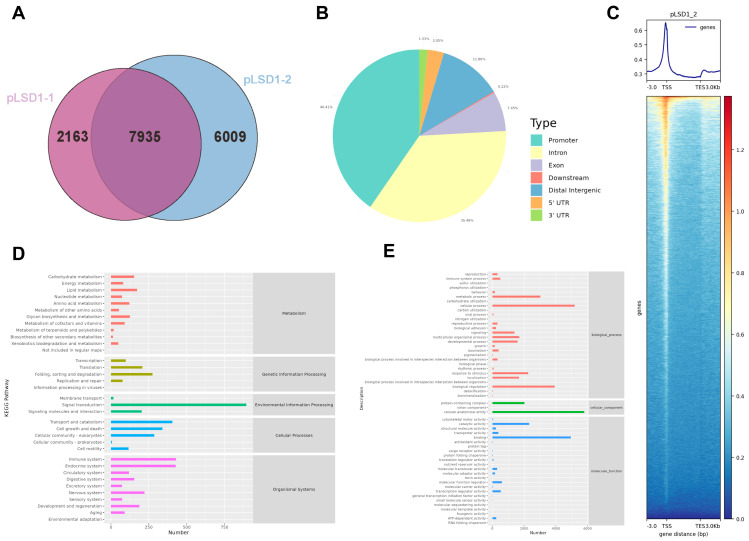
CUT&RUN sequencing analysis of LSD1Ser54p protein. (**A**) Venn plot of LSD1Ser54p showing number of similarities and differences in annotated genes between two sequencing samples. (**B**) Pie charts of different functional components enriched in peaks. (**C**) Average signal value of all genes at 3 kb upstream and downstream of transcription start site (TSS). Annotated target gene of LSD1Ser54p was used for KEGG enrichment analysis (**D**) and GO enrichment analysis (**E**).

**Figure 6 biomolecules-14-01343-f006:**
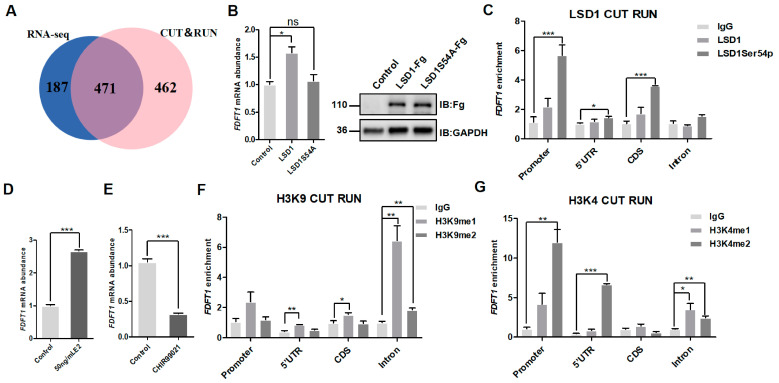
An analysis of target genes coregulated by estrogen and LSD1Ser54p. (**A**) The Venn diagram shows the intersection genes of RNA-seq and CUT&RUN sequencing. (**B**) LSD1 and LSD1S54A plasmids were transfected into chicken Post-TCs, and after 24 h, total RNA and protein were extracted for qPCR and Western blotting. The right figure shows the transfection efficiency. (**C**) In chicken Post-TCs, CUT&RUN was performed with the corresponding antibodies, and DNA was extracted for a qPCR assay. (**D**) Chicken Post-TCs were treated with 50 ng/mL estrogen, and total RNA was extracted for a qPCR assay 24 h later. (**E**) Chicken Post-TCs were treated with 5 μmol of CHIR99021 for 24 h; total RNA was extracted for qPCR detection. In chicken Post-TCs, CUT&RUN was performed with the corresponding antibodies (**F**) H3K9me1 and H3K9me2 and (**G**) H3K4me1 and H3K4me2, DNA was extracted for qPCR testing. * *p* < 0.05, ** *p* < 0.01, *** *p* < 0.001, ^ns^
*p* > 0.05.

**Figure 7 biomolecules-14-01343-f007:**
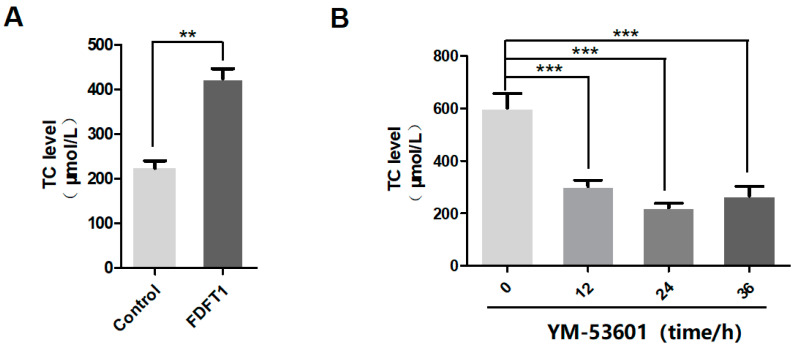
Regulation of cholesterol production by *FDFT1.* (**A**) *FDFT1* was overexpressed in chicken Post-TCs, and changes in cholesterol levels were measured by ELISA. (**B**) Chicken Post-TCs were treated with 1 μmol of YM-53601; changes in cholesterol levels at 12, 24, and 36 h were detected by ELISA. ** *p* < 0.01, *** *p* < 0.001.

**Figure 8 biomolecules-14-01343-f008:**
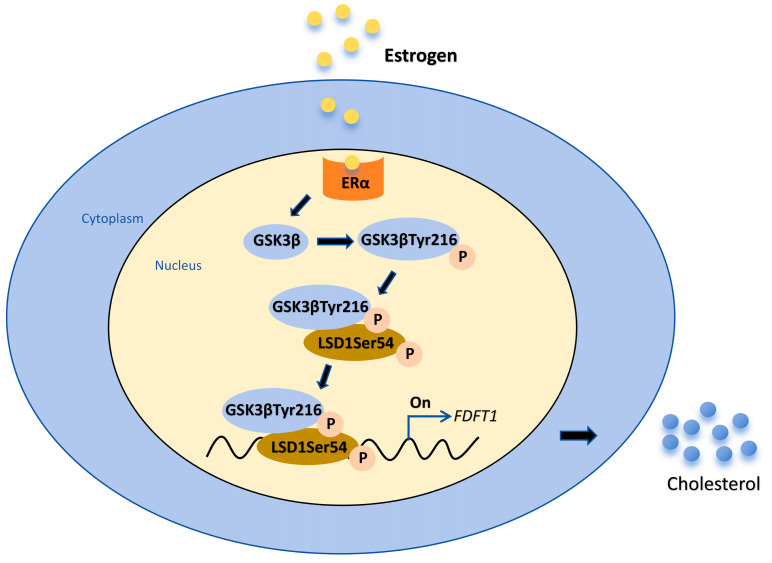
Schematic diagram of regulatory mechanism of GSK3β/LSD1 before and after estrogen treatment of chicken Post-TCs. Estrogen promotes LSD1Ser54p by increasing GSK3βTyr216p levels, thereby promoting *FDFT1* gene expression and cholesterol production.

## Data Availability

The transcriptome data were deposited in the NCBI Sequence Read Archive (https://www.ncbi.nlm.nih.gov/sra/PRJNA1140136, accessed on 25 July 2024) under accession number PRJNA1140136. The CUT&RUN Sequencing data were deposited in the NCBI Sequence Read Archive (https://www.ncbi.nlm.nih.gov/sra/PRJNA1141760, accessed on 25 July 2024) under accession number PRJNA1141760.

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
