# Peer review of "Estrogen Enhances FDFT1 Expression in Theca Cells of Chicken Hierarchical Ovarian Follicles by Increasing LSD1Ser54p Level Through GSK3β Phosphorylation at 216th Tyrosine"

_biomolecules, 2024, doi:10.3390/biom14111343_

Round 1
Reviewer 1 Report
Comments and Suggestions for Authors
The manuscript can be accepted after done this correction
1. Throughout the manuscript article needs grammar correction
2. The images needs to be improved for all the manuscript
3. The novelty of your work should be mentioned in the introduction part
4. Cite the figure in the respective place
5. The passage number of the cell line should be mentioned
6. Mention the information about the chemicals and instrument details
7. Need to correct spacing error
8. The letters were not followed uniform size
9. How was the fluorescent in the cells experiment was calculated
10. The recent reference article to support this work was missing. Cite this article: https://doi.org/10.1007/s11033-022-07544-5; https://doi.org/10.1016/j.jksus.2021.101665; https://doi.org/10.1016/j.scitotenv.2022.160968; https://doi.org/10.1016/j.cbpc.2022.109463; https://doi.org/10.1007/s10989-022-10395-0
The languages needs to be improved
Reviewer 2 Report
Comments and Suggestions for Authors
The article titled "Estrogen Enhances FDFT1 Expression in the Theca Cells of Chicken Hierarchical Ovarian Follicles by Increasing LSD1Ser54p Level through GSK3β Phosphorylation at 216th Tyrosine", focuses on the role of hormones and intracellular signals in regulating follicular development in chickens and the synthesis cholesterol, which is a key component for the synthesis of steroid hormones. The research findings present a new mechanism by which estrogen may affect cholesterol synthesis.
In my opinion the article is quite important because it fills the research gap that exists on the molecular processes behind the regulation of cholesterol synthesis and follicular development in chickens. The article could be published if some minor issues are addressed:
-I suggest that figure 8 be recreated, to better show the process of GSK3β phosphorylation and the binding to LSD1Ser54p.
-A table summarizing the key proteins and genes involved with a brief description of their role could be introduced into the discussion.
Reviewer 3 Report
Comments and Suggestions for Authors
In this study, the role of estrogen in the development of chicken ovarian follicles was examined. Previous findings have shown that estrogen increases the phosphorylation level of LSD1 at serine 54 (LSD1Ser54p) in the theca cells of chicken follicles. The authors noted that GSK3β directly interacts with LSD1 to increase the levels of LSD1Ser54p. Additionally, they demonstrated that the expression of the farnesyl-diphosphate farnesyltransferase 1 (FDFT1) gene is regulated by the effects of estrogen and GSK3β on LSD1Ser54p, and that the overexpression of FDFT1 enhances cholesterol biosynthesis in chicken theca cells. Consequently, the study provides evidence that the effect of estrogen on GSK3β increases the expression of LSD1Ser54p and FDFT1, thereby supporting cholesterol biosynthesis, which in turn enhances estrogen synthesis. Furthermore, the authors have presented the study in a clear, fluent style with good organization.
Reviewer comments and recommendation are listed below:
-Lines 18 and 73, the abbreviation for Glycogen Synthase Kinase 3 Beta is introduced twice. Please revise accordingly.
-Line 34, provide more details on how follicles are prone to atresia for better understanding.
-Extra spaces should be removed (e.g. line 49).
-Line 97, the statement in parentheses should be written with a space.
-Line 198, species names must be italicized.
-The discussion section has been explained in sufficient detail, but the future direction and potential contributions of the study could be elaborated in more detail based on the literature.
-The potential impacts could be given in a conclusions section, which is missing and could help the readers to cover the general aspects of the study.
Round 2
Reviewer 3 Report
Comments and Suggestions for Authors
the authors made all the necessary changes to improve the quality of the paper